# MicroRNA29a Reverts the Activated Hepatic Stellate Cells in the Regression of Hepatic Fibrosis through Regulation of ATPase H^+^ Transporting V1 Subunit C1

**DOI:** 10.3390/ijms20040796

**Published:** 2019-02-13

**Authors:** Fei Jing, Yan Geng, Xin-Yi Xu, Hong-Yu Xu, Jin-Song Shi, Zheng-Hong Xu

**Affiliations:** 1School of Pharmaceutical Science, Jiangnan University, Wuxi 214122, China; 6161502006@vip.jiangnan.edu.cn (F.J.); 6171504010@stu.jiangnan.edu.cn (X.-Y.X.); shijs@163.com (J.-S.S.); 2National Engineering Laboratory for Cereal Fermentation Technology, Jiangnan University, Wuxi 214122, China; xhyhyhy@163.com; 3The Key Laboratory of Industrial Biotechnology, Ministry of Education; School of Biotechnology, Jiangnan University, Wuxi 214122, China

**Keywords:** cell differentiation, fibrosis regression, hepatic fibrosis, peroxisome proliferators-activated receptors gamma (PPARγ), vacuolar adenosine triphosphatase (V-ATPase)

## Abstract

Activated hepatic stellate cells (aHSCs) play a key role in liver fibrosis. During the regression of fibrosis, aHSCs are transformed into inactivated cells (iHSCs), which are quiescent lipid-containing cells and express higher levels of lipid-related genes, such as peroxisome proliferators-activated receptors gamma (PPARγ). Here, we investigated the role of *MicroRNA29a (Mir29a)* in the resolution of liver fibrosis. *Mir29a* and lipid-related genes were up-regulated after the recovery of CCl_4_-induced liver fibrosis in mice. PPARγ agonist rosiglitazone (RSG) promoted de-differentiation of aHSCs to iHSCs and up-regulated *MIR29a* expression in a human HSC cell line LX-2. *MIR29a* mimics in vitro promoted the expression of lipid-related genes, while decreased the expression of fibrosis-related genes. *MIR29a* inhibitor showed the reverse effects. ATPase H^+^ transporting V1 subunit C1 (Atp6v1c1) was increased in liver fibrosis, while down-regulated after the recovery in mice, and negatively regulated by *MIR29a* in LX-2 cells. Knockdown of ATP6V1C1 by siRNA decreased alpha-smooth muscle actin (α-*SMA*) and increased lipid-related genes expression. Simultaneous addition of *MIR29a* mimics and *ATP6V1C1* siRNA further increased RSG promoted expression of lipid-related proteins in vitro. Collectively, *MIR29a* plays an important role during the trans-differentiation of aHSCs in the resolution of liver fibrosis, in part, through regulation of *ATP6V1C1*.

## 1. Introduction

Fibrosis diseases cause serious harm to human health [1], such as pulmonary fibrosis, cirrhosis, glomerulosclerosis, myelofibrosis, and hypertrophic scars [2]. Liver fibrosis is the liver’s wound healing response to a variety of external injuries, which can cause cirrhosis and even hepatocellular carcinoma. However, there is no effective treatment currently [3]. More and more evidence showed that the process of liver fibrosis is reversible after the removal of pathogenic factors [4,5,6]. So, the study of the molecular mechanisms in the resolution of liver fibrosis may be essential for prevention and treatment.

Quiescent hepatic stellate cells (qHSCs) are present in normal liver and store retinoids in large amounts of lipid droplets [7]. Under pathological conditions, qHSCs are activated by cytokines released by immune cells and hepatocytes, and gradually lose their retinoid-filled lipid droplets [8]. At the same time, the synthesis and deposition of extracellular matrix (ECM), including collagen type I (*Col1*), are increased [9,10]. Genetic tracing studies revealed that HSCs are the main ECM producer during hepatic fibrogenesis [11,12].

Activated hepatic stellate cells (aHSCs) may revert to inactivated HSC (iHSCs) in vivo and in vitro [13,14,15]. iHSCs and qHSCs are both rich in lipid droplets and have lipofibroblast properties. iHSCs highly expressed lipid-related genes, such as peroxisome proliferators-activated receptors gamma (*PPARγ*), and lowly expressed fibrotic-related genes, such as alpha-smooth muscle actin (*α-SMA*) and *COL1*. However, iHSCs can differentiate into myofibroblasts faster than qHSCs in response to fibrogenic stimuli [5,6]. It is well known that *PPARγ* signaling plays an important role in the trans-differentiation of myofibroblasts to lipofibroblast. Overexpression of *PPARγ* promotes the de-differentiation of aHSCs into iHSCs in vitro [16]. PPARγ agonist rosiglitazone (RSG) can inhibit ECM production and may serve as potential therapeutics for lung fibrosis and intestinal fibrosis [17,18]. The above studies indicate that the trans-differentiation of myofibroblast to lipofibroblast may play a role in reversing fibrosis.

MicroRNAs (miRNAs) are non-encoding single-stranded RNA molecules of about 22 nucleotides in length encoded by endogenous genes [19]. They are involved in post-transcriptional gene expression regulation in plants and animals [20]. The *MIR29* family has been reported to participate in the development of cardiac fibrosis, liver fibrosis, renal fibrosis, and pulmonary fibrosis [21,22,23,24]. *MIR29* has three mature members, *MIR29a*, *MIR29b,* and *MIR29c*. *MiR29* is significantly under-expressed in human and murine fibrotic liver tissues [25,26]. Overexpression of *Mir29* in mouse HSCs leads to a decrease in collagen deposition through direct target ECM production [27]. Patients with advanced cirrhosis have a significantly lower serum level of *MIR29a* compared with healthy volunteers or patients with early hepatic fibrosis [28]. *Mir29a* also influences the expression of genes associated with lipid metabolism in mouse C2C12 myoblasts [29,30]. These studies suggest that enhancing *MIR29* may be a promising anti-fibrotic therapy. However, the mechanism of action of *MIR29a* in liver fibrosis remains largely unclear.

Vacuolar adenosine triphosphatase (V-ATPase) has been shown to play an important role in the maintenance of the intracellular pH. V-ATPase is composed of a cytosolic V1 domain and a transmembrane V0 domain. The V1 domain consists of three A subunits, three B subunits, two G subunits plus the C, D, E, F, and H subunits. V-ATPase inhibitor affects the proliferation, activation, and metabolic activity of HSC [31]. *ATP6V1C1*, one of the main genes regulating V-ATPase activity, has been implicated in the regulation of energy metabolism [32,33]. ATP6V1C1 expression is up-regulated in the activated primary human HSC and co-localized with HSC activation marker α-SMA in human liver tissues of patients with cirrhosis [34].

In this study, we focused on investigating the role of *MIR29a* in the resolution of liver fibrosis. Mouse liver fibrosis model induced by CCl_4_ and human HSC cell line LX-2 were used. We hypothesized that *MIR29a* promotes the resolution of liver fibrosis, and *ATP6V1C1* may be a potential target of *MIR29a*.

## 2. Results

### 2.1. Expression of Mir29a, Fibrogenic- and Adipogenic-Related Genes in the Progression and Resolution of Liver Fibrosis in Mice

First of all, we established the liver fibrosis model by repetitive CCl_4_ injection in mice. By quantitative real-time PCR (qRT-PCR), we found that *α-Sma* and *Col1* were significantly up-regulated (Figure 1A,B), while the expression of *Mir29a* was decreased in liver tissues after 28 days with CCl_4_ treatment, as compared to controls (CTL) (Figure 1C). Meanwhile, the adipogenic transcription factors *Pparγ,* adipose differentiation-related protein (*Adrp*)*,* and sterol regulatory element binding transcription factor 1 (*Srebp-1c*) were down-regulated in liver tissues after 28 days of CCl_4_ treatment, as compared to CTL (Figure 1D–F). After 7 days of recovery from CCl_4_, the expression of *Col1* was down-regulated (Figure 1B), while mRNA levels of *Mir29a*, *Pparγ*, and *Srebp-1c* were increased in liver tissues compared with those of the model group (Figure 1C,D,F). The mRNA levels of *α-Sma* and *Col1* were significantly reduced 30 days after recovery (Figure 1A,B), indicating that the degree of liver fibrosis was greatly reduced. The expression of *Mir29a* was increased (Figure 1C), and mRNA levels of *Pparγ*, *Adrp*, and *Srebp-1c* were similar in the liver after 30 days of recovery compared to the CTL group (Figure 1D–F). These data indicated that *Mir29a* is negatively correlated with the fibrosis progress and may play a role in the resolution of liver fibrosis.

### 2.2. The Effect of MIR29a on aHSCs Trans-Differentiation to iHSCs

PPARγ plays a key role in the reversal of aHSC to iHSC. We found that PPARγ agonist RSG (1.25–80 μM) showed no significant effect on cell viability (Appendix A). Treatment with RSG (5 μM) resulted in a dramatic up-regulation of *PPARγ* expression in LX-2 cells (Appendix A). RSG up-regulated *MIR29a* expression in LX-2 cells (Appendix A). Instead, mRNA levels of both *COL1* and *α-SMA* in LX-2 cells were down-regulated by RSG (Appendix A). We also confirmed that RSG induced the adipogenic related genes (PPARγ, ADRP, FASN, C/EBPα) expression and down-regulated α-SMA expression in protein levels (Appendix A). Then, to determine whether *MIR29a* plays a role in the aHSC trans-differentiation, we used *MIR29a* mimics or *MIR29a* inhibitor in LX-2 cells treated with RSG. The results of Oil red O staining showed that *MIR29a* mimics or RSG induced lipid droplets formation in LX-2 cells (Figure 2A,B). The quantitative data revealed that *MIR29a* mimics further promoted RSG-induced lipid accumulation (Figure 2B). *MIR29a* expression was higher in LX-2 cells treated with RSG (Figure 2C). *MIR29a* mimics alone decreased the expression of the fibrosis-related genes *COL1* and *α-SMA* compared to the CTL group (Figure 2D,E). Meanwhile, overexpression of *MIR29* with its mimics resulted in 1.7, 1.3, 1.5, and 1.4 fold up-regulation of *PPARγ*, *ADRP*, *FAS*, and *SREBP-1c* expression in LX-2 cells, respectively (Figure 2F–I). Furthermore, *PPARγ*, *ADRP*, *FASN*, and *SREBP-1c* were increased all above 1.5 fold upon treatment with both RSG and *MIR29a* mimics (Figure 2F–I). Conversely, *MIR29a* inhibitor increased *COL1* and *αSMA* (Appendix A) but decreased *PPARγ*, *ADRP*, *FASN*, and *SREBP-1c* expression (Appendix A). These results suggested that *MIR29a* may promote aHSC trans-differentiation.

### 2.3. The Effect of ATP6V1C1 on aHSCs Trans-Differentiation to iHSCs

To investigate the potential new target of *MIR29a*, we used the bioinformatics analysis and found that *ATP6V1C1* might be the potential target of *MIR29a* [35]. We first examined the expression of *Atp6v1c1* in murine liver tissues. The mRNA level of *Atp6v1c1* in liver tissues was increased more than two-fold after 28 days with CCl_4_ treatment and was again similar to that of CTL after fibrosis recovery (Figure 3A). We also found that *MIR29a* negatively regulated the expression of ATP6V1C1 by qRT-PCR and western blotting analyses (Figure 3B–D).

We then examined whether the reduction of *ATP6V1C1* by siRNA can play a role in the HSC trans-differentiation. As expected, *ATP6V1C1* siRNA effectively decreased *ATP6V1C1* expression relative to a scramble siRNA control in LX-2 cells (Figure 4A). Reduction of *ATP6V1C1* by siRNA alone led to the inhibition of *α-SMA* (Figure 4B) and up-regulation of *PPARγ*, *ADRP*, and *FASN* (Figure 4C–E). In addition, RSG slightly decreased *ATP6V1C1* expression (Figure 4A). *ATP6V1C1* siRNA further promoted the expression of *PPARγ*, *ADRP*, *FASN* induced by RSG (Figure 4C–E). The results of Oil red O staining showed that knockdown of *ATP6V1C1* results in a 1.5 fold up-regulation of *MIR29a* mimics and RSG-induced lipid differentiation (Figure 4F,G). At the protein level, knockdown of *ATP6V1C1* increased the expression of PPARγ, ADRP, and FASN in LX-2 cells treated with *MIR29a* mimics and RSG (Figure 4H). These data suggested that *ATP6V1C1* inhibits the HSC lipogenesis and may be the target of *MIR29a* in the progress of HSC trans-differentiation.

## 3. Discussion

This work highlights the importance of *MIR29a* during the regression of liver fibrosis. *MiR29a* has been known as a direct target of TGFβ signaling and is down-regulated when liver fibrosis occurs. Overexpressing *Mir29a/b* markedly reduced the degree of liver fibrosis induced by CCl_4_ in mice and decreased Collagen expression in LX-2 cells [28,36]. Ectopic expression of *Mir29b* also blunted the increased expression of *α-SMA*, caused cell cycle arrest, and induced apoptosis in activated HSCs [37]. Liver-specific *Mir29ab1* knockout mice enhanced mortality and fibrosis in response to CCl_4_ treatment compared with wild-type CTL mice [38]. Consistent with these studies, our study found that *Mir29a* was down-regulated in CCl_4_-induced liver fibrosis in mice. We further showed that the expression level of *Mir29a* in the liver was also recovered and comparable to that of normal mice after 30 days of recovery from CCl_4_ treatment. In vitro, *MIR29a* mimics significantly decreased the expression of *COL1* and *α-SMA* in human LX-2 cells.

A treatment of aHSC with ectopic expression of *PPARγ* induced the expression of a panel of adipogenic transcription factors and caused the phenotypic reversal to qHSC [16]. In vitro study also found that *Mir29a* promotes lipid production in mouse skeletal muscle cells [29,30]. As expected, we found that RSG increased the expression of *PPARγ* and induced lipid droplet formation in LX-2 cells. Moreover, our work showed that RSG up-regulated *MIR29a* expression. Meanwhile, *MIR29a* promoted lipid droplet formation and up-regulated the expression of *PPARγ* and several adipogenic transcription factors. These data suggest that *MIR29a* plays an important role in the process of aHSCs to iHSCs trans-differentiation during liver fibrosis resolution.

The previous study reported that V-ATPase, including ATP6V1C1, was co-localized with HSC activation marker α-SMA in human cirrhotic liver tissues determined by immunohistochemistry analysis, and pharmaceutical inhibition of V-ATPase led to the down-regulation of gene expression of HSC activation markers [34]. In this study, by bioinformatics analysis, we revealed that *ATP6V1C1* might be the target of *MIR29a* [35]. The expression of *Atp6v1c1* was negatively correlated with *Mir29a* in CCl_4_-induced liver fibrosis in mice as well as in human HSC in vitro. Next, we confirmed that ATP6V1C1 was negatively regulated by *Mir29a* in LX-2 cells. ATP6V1C1 is a potent negative regulator of HSC trans-differentiation. Knockdown of *ATP6V1C1* led to the inhibition of *α-SMA*, whereas increased the expression of the adipogenic transcription factors, including *PPARγ*, *ADRP*, and *FASN*. ATP6V1C1 siRNA further increased lipid deposition and the expression of lipid-related protein stimulated by *MIR29a* mimics and RSG. Thus, *ATP6V1C1* may be regulated by *MIR29a* and promote the fibrogenesis while inhibiting the lipogenesis progress during liver fibrosis.

## 4. Materials and Methods

### 4.1. Chemicals and Reagents

Carbon tetrachloride (CCl_4_), olive oil, oil red O, and RSG were from Sigma-Aldrich (St Louis, MO, USA). The mimics and inhibitor of *MIR29a* were purchased from the Ruibo Company (Guangzhou, China). ATP6V1C1 siRNA and scramble siRNA control were purchased from Santa Cruz Biotechnology (Santa Cruz, CA, USA). The antibodies ATP6V1C1, PPARγ, ADRP, FASN, and GAPDH were purchased from Santa Cruz Biotechnology (Santa Cruz, CA, USA).

### 4.2. Animal Model of Liver Fibrosis and Treatment

All animal experiments were approved by the Animal Research Committee of Jiangnan University (JN.No 20171115c0400424[72]; 2017.12.24–2018.4.24). Male C57LB/6 at 8 weeks were purchased from Shanghai SLAC Laboratory Animal CO. LTD and were allowed to freely drink tap water and diet (Shanghai SLAC Laboratory Animal Co., Shanghai, China). Liver fibrosis was induced by intraperitoneal (i.p.) injection of 0.5 mL/kg CCl_4_ (25% solution in olive oil) twice weekly for 28 days [39]. The mouse livers were harvested at 7 and 30 days after CCl_4_ injury (*n* = 5 per group).

### 4.3. Cell Culture, Cell Viability, and Drug Treatment

Human HSCs cell line LX-2 cells were obtained from the cell bank of Xiangya Central Experiment Laboratory of Central South University (Changsha, China). Cells were cultured in DMEM supplemented with 10% FBS (Gibco, Grand Island, NY, USA), 100 U mL^−1^ penicillin, and 100 mg mL^−1^ streptomycin in a humidified atmosphere of 5% CO_2_ at 37 °C. *MIR29a* mimics (100 nM), *MIR29a* inhibitor (200 nM), ATP6V1C1 siRNA (60 nM) or scramble siRNA control (60 nM) were transiently transfected into LX-2 cells for 24 h using Lipofectamine RNAi max (Invitrogen, Carlsbad, CA, USA). The cells were then treated with 5 µM RSG for 24 h.

### 4.4. RNA Isolation and qRT-PCR Analysis

Trizol reagent (Invitrogen, Carlsbad, CA, USA) was used for extracting RNA from mouse liver tissues or cells according to the manufacturer’s protocol. Total RNA (1 mg) was transcribed into cDNA using M-MLV and random hexamer primers (Thermo Fisher Scientific, Waltham, MA, USA). Real-time PCR was performed as previously described [40]. *Glycerol phosphate dehydrogenase* (*GAPDH*) was used as an invariant control. Gene expressions were measured relative to the endogenous reference gene *GAPDH* using the comparative CT method. Sequences of the specific primer sets are as follows: *Gapdh* (NM-008085), forward, 5’-TTG TCA TGG GAG TGA ACG AGA-3’; reverse, 5’-CAG GCA GTT GGT GGT ACA GG-3’; *GAPDH* (NM-014364), forward, 5’-TGT GGG CAT CAA TGG ATT TGG-3’; reverse, 5’-ACA CCA TGT ATT CCG GGT CAA T-3’; *Col1* (NM-007742), forward, 5’-GCT CCT CTT AGG GGC CAC T-3’; reverse, 5’-ATT GGG GAC CCT TAG GCC AT-3’; *COL1* (NM-000088), forward, 5’-GAG GGC CAA GAC GAA GAC ATC-3’; reverse, 5’-CAG ATC ACG TCA TCG CAC AAC-3’; *α-Sma* (NM-007392), forward, 5’-CCC AGA CAT CAG GGA GTA ATG G-3’; reverse, 5’-TCT ATC GGA TAC TTC AGC GTC A-3’; *α-SMA* (NM-001613), forward, 5’-GTG TTG CCC CTG AAG AGC AT-3’; reverse, 5’-GCT GGG ACA TTG AAA GTC TCA-3’; *Pparγ* (NM-001308352), forward, 5’-CTT GGC TGC GCT TAC GAA GA-3’; reverse, 5’-GAA AGC TCG TCC ACG TCA GAC-3’; *PPARγ* (NM-001172698), forward, 5’-GAT GCC AGC GAC TTT GAC TC-3’; reverse, 5’-ACC CAC GTC ATC TTC AGG GA-3’; *Adrp* (NM-007408), forward, 5’-CTT GTG TCC TCC GCT TAT GTC-3’; reverse, 5’-GCA GAG GTC ACG GTC TTC AC-3’; *ADRP* (NM-001122), forward, 5’-ATG GCA TCC GTT GCA GTT GAT-3’; reverse, 5’-GGA CAT GAG GTC ATA CGT GGA G-3’; *Srebp-1*c (NM-011480), forward, 5’-TGA CCC GGC TAT TCC GTG A-3’; reverse, 5’-CTG GGC TGA GCA ATA CAG TTC-3’; *SREBP-1C* (NM-001005291), forward, 5’-GCC CCT GTA ACG ACC ACT G-3’; reverse, 5’-CAG CGA GTC TGC CTT GAT G-3’; *FASN* (NM-012393), forward, 5’-CCC AGT CCT TCA CTT CTA TGT TC-3’; reverse, 5’-GTA GCA CAG TTC AGT CTC GAC-3’. The expression level of mature *MIR29a* was quantified by TaqMan microRNA assays (Mm04238191_s1, Applied Biosystems, Foster city, CA, USA).

### 4.5. Western Blot Analysis

Cells were washed with ice-cold Dulbecco’s Phosphate Buffered Saline (Gibco, Grand Island, NY, USA) and resuspended in RIPA buffer with protease inhibitor (Sigma-Aldrich, St.Louis, MO, USA). Protein was resolved by SDS-polyacrylamide gel electrophoresis and transferred to PVDF membranes. After blocking, they were probed with primary antibodies overnight at 4 °C, then incubated with horseradish peroxidase-conjugated secondary antibody for 1 h at room temperature. The bands were visualized using ECL reagents (Thermofisher Scientific, Waltham, MA, USA). The ratio of the relevant protein was subjected to internal control (GAPDH).

### 4.6. Oil Red O Staining

Oil Red O solution 0.5% in isopropanol was used to stain fat droplets in cells. After removing the staining solution, the dye retained in the cells was eluted into isopropanol, and the optical density (OD) at 510 nm was measured with a microplate reader (Thermo Labsystems, Waltham, MA, USA).

### 4.7. Statistical Analysis

Data are expressed as means ± SEM. Differences in measured variables between experimental and control groups were assessed by using Student’s test. Differences in multiple groups were assessed by using one-way analysis of variance (ANOVA), and the Tukey test was used for determining the significance. Results were considered statistically significant at *p* < 0.05. All analyses were conducted in GraphPad Prism (La Jolla, CA, USA).

## 5. Conclusions

Our study found that *Mir29a* and the adipogenic transcription factors were up-regulated during the process of fibrosis regression in mice. *MIR29a* down-regulated the expression of *ATP6V1C1*, inhibited the activation of HSCs, and positively regulated the adipogenic transcription factors. *ATP6V1C1* negatively regulated adipogenic transcription factors. We propose a new mechanism of action of *MIR29a* in the resolution of liver fibrosis. *MIR29a* plays an important role during the resolution of liver fibrosis, which may be through the regulation of *ATP6V1C1*.

## Figures and Tables

**Figure 1 ijms-20-00796-f001:**
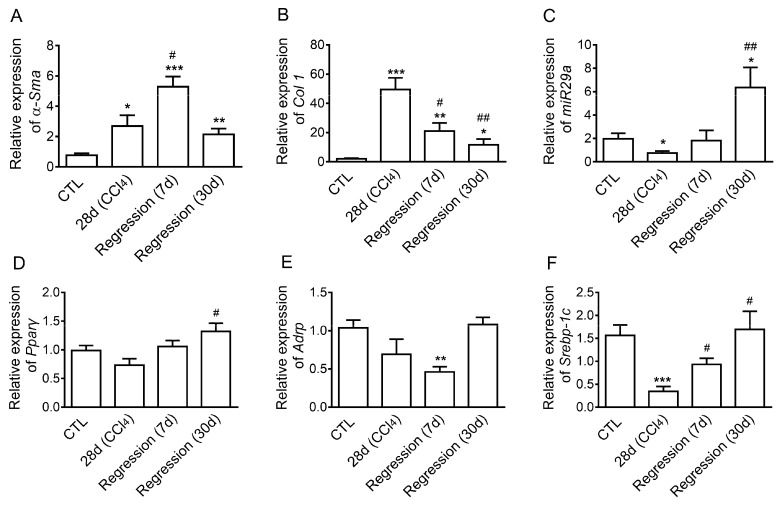
Expression of *Mir29a* and adipogenic-related genes expression increased, fibrogenic-related genes expression decreased in the resolution of liver fibrosis in mice. (**A**–**F**) Mice and littermate control mice were treated with oil or CCl_4_ for 28 days, and liver tissues were harvested for the following analyses at the indicated time points. qRT-PCR analysis of *α-Sma*, *Col1*, *Mir29a*, *Pparγ*, *Adrp*, and *Srebp-1c* mRNA expression (*n* = 5 per group). Error bar represents SEM. * *p* < 0.05, ** *p* < 0.01, and *** *p* < 0.001 vs. oil group. ^#^
*p* < 0.05, ^##^
*p* < 0.01 vs. CCl_4_ group.

**Figure 2 ijms-20-00796-f002:**
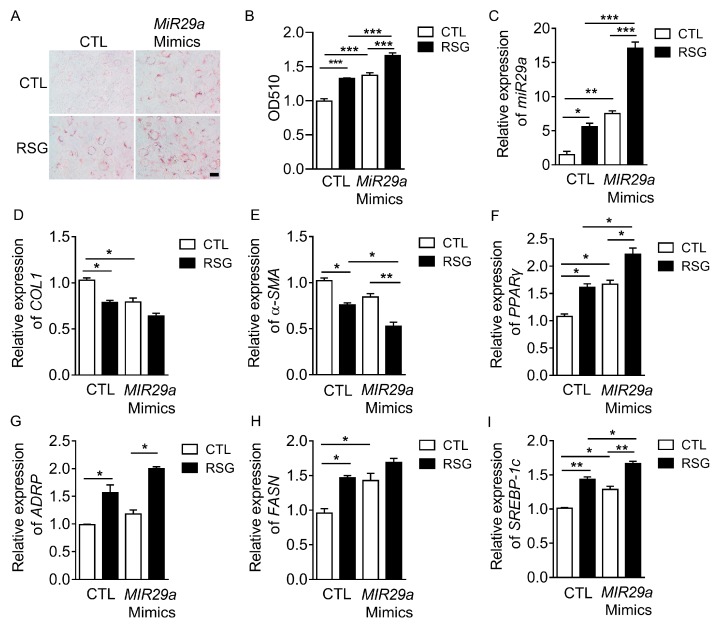
*MIR29a* mimics decreased the expression of fibrosis-related genes and increased the expression of adipogenic transcription factors. (**A**,**B**) *MIR29a* mimics (100 nM) were transiently transfected into LX-2 cells for 48 h, and the cells were treated with RSG (5 µM) for 24 h. Conventional Oil red O staining was conducted. After removing the staining solution, the dye retained in the cells was eluted into isopropanol, and OD510 was determined. Scale bar: 10 µm. (**C–I**) The expression of *MIR29a*, *COL1*, *α-SMA*, *PPARγ*, *ADRP*, *FASN*, and *SREBP-1c* mRNA were assessed by qRT-PCR (*n* = 3 per group). Error bar represents SEM. * *p* < 0.05, ** *p* < 0.01, and *** *p* < 0.001.

**Figure 3 ijms-20-00796-f003:**
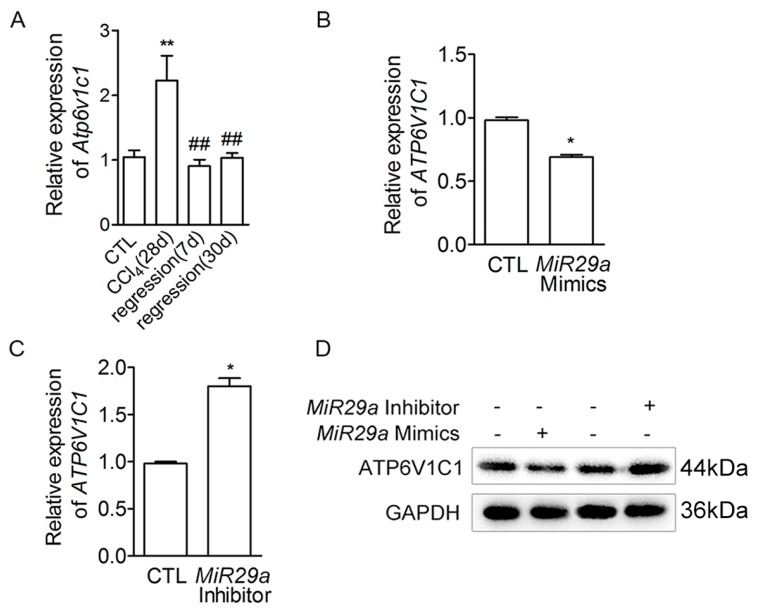
The expression of *ATP6V1C1* in cells and CCl_4_-induced mice. (**A**) Mice and littermate control mice were treated with oil or CCl_4_ for 28 days, and liver tissues were harvested for the following analyses at the indicated time points. qRT-PCR analysis of *Atp6v1c1* mRNA expression (*n* = 5 per group). Error bar represents SEM. ** *p* < 0.01 vs. CTL group. ^##^
*p* < 0.01 vs. CCl4 (28d) group. (**B**,**C**) *MIR29a* mimics (100 nM) or *MIR29a* inhibitor (200 nM) were transiently transfected into LX-2 cells for 24 h, and the cells were treated with RSG (5 µM) for 24 h. The expression of *ATP6V1C1* mRNA was assessed by qRT-PCR (*n* = 3 per group). Error bar represents SEM. * *p* < 0.05. (**D**) Protein expression levels of ATP6V1C1 in cell extracts were assessed by Western blot. GAPDH was used as a loading control.

**Figure 4 ijms-20-00796-f004:**
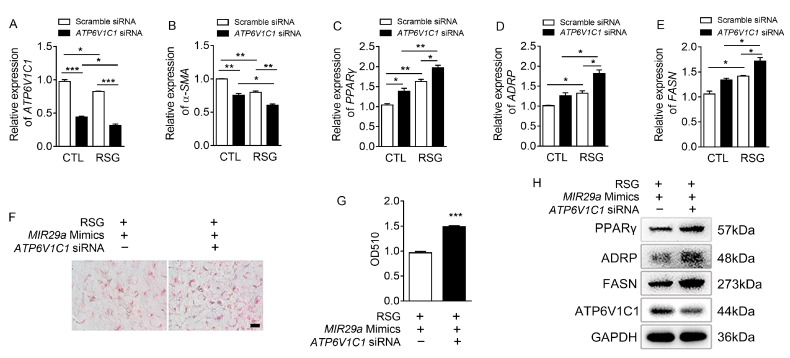
Knockdown of *ATP6V1C1* attenuated the activation of LX-2 cells and up-regulated the expression of the adipogenic transcription factors. (**A**–**E**) ATP6V1C1 siRNA (60 nM) transfection was performed on LX-2 cells using Lipofectamine RNAimax. The gene expressions of *ATP6V1C1*, *α-SMA*, *PPARγ*, *ADRP,* and *FASN* mRNA were assessed by qRT-PCR (*n* = 3 per group). (**F**,**G**) *MIR29a* mimics (100 nM) and ATP6V1C1 siRNA (60 nM) were transiently transfected into LX-2 cells for 24 h, and the cells were treated with RSG (5 µM) for 24 h. Conventional Oil red O staining was conducted. After removing the staining solution, the dye retained in the cells was eluted into isopropanol, and OD510 was determined. Scale bar: 10 μm. (**H**) Protein expression levels of PPARγ, ADRP, FASN, and ATP6V1C1 in cell extracts were assessed by Western blot. GAPDH was used as a loading control. Throughout, the error bar represents SEM. * *p* < 0.05, ** *p* < 0.01, and *** *p* < 0.001.

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
