# Peer review of "MicroRNA29a Reverts the Activated Hepatic Stellate Cells in the Regression of Hepatic Fibrosis through Regulation of ATPase H+ Transporting V1 Subunit C1"

_ijms, 2019, doi:10.3390/ijms20040796_

Reviewer 1 Report

The manuscript by Jing et al. describes the role of miR29a in the regression of liver fibrosis. This study is well-designed but has some concerns that need to be addressed:

The manuscript should be be edited and improved for English language by a native English speaker.

Higher resolution images should be presented for the histologies to clearly see the differences in groups.

Are all the Western blot proteins measured on the same blot? If not, please provide the respective GAPDH bands for each blot and also show the molecular weight in the figures.

Some controls in Figure 4 are missing.

The discussion section is redundant and just states that results. Needs to be re-written.

Author Response

We appreciate the reviewer’s expert insights, comments, and effort.

- the manuscript should be edited and improved for English language by a native English speaker.

Response: The manuscript has been edited and improved for English language by a native English speaker.

- higher resolution images should be presented for the histologies to clearly see the differences in groups.

Response: Thanks for the reviewer’s suggestion and we have increased the resolution of the images in the revised manuscript.

- are all the Western blot proteins measured on the same blot? If not, please provide the respective GAPDH bands for each blot and also show the molecular weight in the figures.

Response: Yes. All the Western blot proteins measured on the same blot. And we also showed the molecular weight in the revised figures.

- Some controls in Figure 4 are missing.

Response: Sorry for the misunderstanding. We have already showed the effect of Mir29a on the RSG induced HSC trans-differentiation in Figure 2. So in Figure 4 (F-H) we only compared whether knockdown ATP6V1C1 by siRNA could affect MIR29a mimics and RSG induced HSC trans-differentiation. Scramble siRNA was used as a control.

- The discussion section is redundant and just states that results. Needs to be re-written.

Response: Thanks for the reviewer’s suggestion and we have shorted and re-written the discussion section.

We tried our best to improve the manuscript and made some changes in the manuscript.  These changes will not influence the content and framework of the paper. And here we did not list the changes but marked in red and blue in revised manuscript.

We appreciate for Reviewers’ warm work earnestly, and hope that the correction will meet with approval.

Once again, thank you very much for your comments and suggestions.

Reviewer 2 Report

The authors provide microRNA29a can promote inactivation of HSC after chronic liver injury by up-regulation of ATPase H+transporting V1 subunit C1. They show very clear data microRNA29a down-regulate hepatic stellate cell activation. The methods look appropriate and their data looks convincing and interesting. The conclusion is impressive. Unraveling the molecular mechanisms of liver fibrogenesis is critical to development of its therapies I think this paper is good for publication in the present form.

Author Response

We appreciate the reviewer's expert insights, comments, and effort.

Round  2

Reviewer 1 Report

The authors have addressed all the concerns. Minor edits in English language grammar are required.

Author Response

We appreciate the reviewer's expert insights, comments, and effort.